# 11p15 Epimutations in Pediatric Embryonic Tumors: Insights from a Methylome Analysis

**DOI:** 10.3390/cancers15174256

**Published:** 2023-08-25

**Authors:** Felipe Luz Torres Silva, Juliana Silveira Ruas, Mayara Ferreira Euzébio, Iva Loureiro Hoffmann, Thais Junqueira, Helder Tedeschi, Luiz Henrique Pereira, Alejandro Enzo Cassone, Izilda Aparecida Cardinalli, Ana Luiza Seidinger, Patricia Yoshioka Jotta, Mariana Maschietto

**Affiliations:** 1Research Center, Boldrini Children’s Hospital, Campinas 13083-884, SP, Brazil; felipeluztorres@outlook.com (F.L.T.S.); ruasjulianas@gmail.com (J.S.R.); ma.euzebio@gmail.com (M.F.E.); jottapaty@gmail.com (P.Y.J.); 2Genetics and Molecular Biology, Institute of Biology, State University of Campinas, Campinas 13083-862, SP, Brazil; 3Boldrini Children’s Hospital, Campinas 13083-210, SP, Brazil

**Keywords:** 11p15, ICR1, ICR2, DNA methylation, embryonic tumors, medulloblastoma, Wilms tumor

## Abstract

**Simple Summary:**

Epigenetics encompasses changes in DNA without changing the DNA sequence itself, and it includes, among other modifications, DNA methylation. DNA methylation consists of the addition of a methyl group (-CH3), usually to a cytosine that precedes a guanine, forming a CpG site, and it is associated with the regulation of gene expression. The disruption of CpG site methylation is widely studied in cancer. This study evaluated the methylation status of a group of CpG sites located at 11p15, which is an imprinted region. There are several imprinted regions in the human genome; they are characterized by being controlled by DNA methylation, allowing only one allele to be expressed. The 11p15 region contains two imprinted control regions (ICR1 and 2) whose changes are considered the underlying mechanism behind Beckwith–Wiedmann syndrome; patients with this syndrome present an increased risk of developing embryonic tumors. Thus, the 11p15 methylation status was evaluated in a series of sporadic embryonic tumors, revealing that a loss of imprinting may happen via the gain or loss of DNA methylation at ICR1 and/or ICR2 and in specific tumor types.

**Abstract:**

Embryonic tumors share few recurrent mutations, suggesting that other mechanisms, such as aberrant DNA methylation, play a prominent role in their development. The loss of imprinting (LOI) at the chromosome region 11p15 is the germline alteration behind Beckwith–Wiedemann syndrome that results in an increased risk of developing several embryonic tumors. This study analyzed the methylome, using EPIC Beadchip arrays from 99 sporadic embryonic tumors. Among these tumors, 46.5% and 14.6% presented alterations at imprinted control regions (ICRs) 1 and 2, respectively. Based on the methylation levels of ICR1 and ICR2, four clusters formed with distinct methylation patterns, mostly for medulloblastomas (ICR1 loss of methylation (LOM)), Wilms tumors, and hepatoblastomas (ICR1 gain of methylation (GOM), with or without ICR2 LOM). To validate the results, the methylation status of 29 cases was assessed with MS-MLPA, and a high level of agreement was found between both methodologies: 93% for ICR1 and 79% for ICR2. The MS-MLPA results indicate that 15 (51.7%) had ICR1 GOM and 11 (37.9%) had ICR2 LOM. To further validate our findings, the ICR1 methylation status was characterized via digital PCR (dPCR) in cell-free DNA (cfDNA) extracted from peripheral blood. At diagnosis, we detected alterations in the methylation levels of ICR1 in 62% of the cases, with an agreement of 76% between the tumor tissue (MS-MLPA) and cfDNA methods. Among the disagreements, the dPCR was able to detect ICR1 methylation level changes presented at heterogeneous levels in the tumor tissue, which were detected only in the methylome analysis. This study highlights the prevalence of 11p15 methylation status in sporadic embryonic tumors, with differences relating to methylation levels (gain or loss), location (ICR1 or ICR2), and tumor types (medulloblastomas, Wilms tumors, and hepatoblastomas).

## 1. Introduction

During normal development, epigenetic modifications are remodeled to define embryo patterning and for organ and cell-type specification; then, upon terminal differentiation, they are maintained to sustain cell identity. When disrupted, the epigenome may play a role in cancer initiation and progression [1]. Several somatic genetic and epigenetic processes contribute to the development of cancer, including copy number alterations, rearrangements, somatic point mutations, and the disruption of DNA methylation [2]. DNA methylation regulates genomic imprinting, which is laid down during the formation of germ cells in the embryonic period [3], determining the expression of either the paternal or the maternal allele [4]. The regions that present differential methylation between DNA alleles, known as CpG islands, are referred to as imprinted control regions (ICRs) [5]. The loss of imprinting (LOI) represents a mutational mechanism in carcinogenesis [5,6]. In patients with a Wilms tumor, for instance, while the matched kidney shows a normal 11p15 methylation status, an LOI can be found in the nephrogenic rests, which are considered precursor lesions [7].

Furthermore, as germline alterations, epimutations increase the risk of developing cancer. Constitutional alterations at 11p15 comprise the mechanisms underlying Beckwith–Wiedemann syndrome (BWS), a pediatric overgrowth disorder that increases the risk of developing embryonic tumors [8,9]. The 11p15 alterations can be detected at mosaic levels, and although these individuals do not necessarily show phenotypic features or a family history of a hereditary cancer, they may have an increased risk of developing cancer, such as Wilms tumors and hepatoblastomas [10,11].

The alterations in 11p15 can occur due to the gain or loss of methylation via the loss of heterozygosity (LOH), uniparental disomy (UPD), 11p15 trisomy, or other rarer alterations [4,12], which may result in an 11p15 LOI [9,12]. Different cancers may exhibit distinct methylation patterns, but abnormalities at 11p15 seem to be relatively common in several tumors. Embryonic tumors that present an 11p15 LOI include Wilms tumors [13,14]; rhabdomyosarcomas, particularly the embryonal subtype [15]; hepatoblastoma [16,17]; and neuroblastomas [18], but there is scarce information on other embryonic tumors, such as Ewing sarcomas and medulloblastomas. Moreover, alterations in imprinting are rarely seen in normal tissues, making the 11p15 epimutation a good candidate for liquid biopsy.

A challenge with pediatric cancers is their low mutational burden with few recurrent mutations [19,20]. However, abnormal DNA methylation is considered an early and frequent event in cancer development, making it a good biomarker to be explored for a targeted methodology. In pediatric tumors, the identification of *RASSF1A* hypermethylation in circulating tumor DNA (ctDNA) showed promising results for neuroblastomas, Wilms tumors, rhabdomyosarcomas, and lymphomas [21]. For Wilms tumors, a hypermethylated region at 6p21.32 was proposed as a biomarker for monitoring tumor response to chemotherapy, as the methylation levels were higher in the cfDNA of patients whose tumors showed a good preoperative chemotherapy response (classified as the regressive subtype) [13].

In this study, we characterized the methylation levels of imprinted control regions (ICRs) 1 and 2, located at 11p15, in sporadic embryonic tumors, including tumors that lacked this information, and tested the ICR1 methylation level as a biomarker for use in liquid biopsy.

## 2. Materials and Methods

### 2.1. Patient Eligibility and Sample Collection

The following primary tumor samples from 99 consecutive patients diagnosed with embryonic tumors were retrieved from the institutional biobank: Wilms tumor (n = 17), neuroblastoma (n = 16), Ewing sarcoma (n = 17), clear cell sarcoma of the kidney (n = 2), hepatoblastoma (n = 3), embryonal rhabdomyosarcoma (n = 4), rhabdomyosarcoma (n = 1), group 3/4 medulloblastoma (n = 12), sonic hedgehog (SHH)-activated medulloblastoma (n = 4), *WNT*-activated medulloblastoma (n = 5), unclassified medulloblastoma (n = 8), *FOXR2*-activated pineoblastoma (n = 1), GRP1A-subgroup pineoblastoma (n = 1), H3 K27-altered diffuse midline glioma (n = 3), pleomorphic xanthoastrocytoma (n = 1), ganglioneuroblastoma (n = 1), atypical teratoid rhabdoid tumor (n = 2), and osteoblastoma (n = 1). Groups of 29 and 6 patients also had their blood collected upon diagnosis and at follow up, respectively. Some of these samples were used previously. DNA and circulating free DNA (cfDNA) extraction methods as well as quality control techniques have been described elsewhere [22].

### 2.2. Determining Tumor Methylation Status via EPIC Beadchip Arrays

DNA extracted from the tumor tissues was subjected to bisulfite conversion using an EZ DNA Methylation Direct (cat D5002, Zymo Research, Irvine, CA, USA). The converted DNA was then hybridized via EPIC Beadchip Methylation arrays (Illumina, San Diego, CA, USA) and scanned using the NextSeq550 (Illumina Inc., San Diego, CA, USA), which generated IDAT files.

The IDAT files were analyzed using the minfi R package, version 1.46.0 [23]. Samples that passed the quality control parameters, as determined via both the BeadArray Controls Reporter software and minfi QCreport function, were normalized using FunNorm [24]. Probes located at the genomic regions ICR1 (GRCh37/hg19 chr11:2019379-2020007) and ICR2 (GRCh37/hg19 chr11:2720540-2721394), as specified in the Infinium MethylationEPIC v1.0 B5 Manifest File, were retrieved and used to construct a heatmap. The R package pvclust was used to assess the uncertainty in the bootstrap hierarchical clustering of the samples [25]. CpG sites were ordered according to their genome location.

The average beta values of ICR1 and ICR2 were compared to MS-MLPA data. Beta-values ranging between 40% and 60% were considered normal, and loss of methylation (LOM) or gain of methylation (GOM) were considered for average beta values below 40% or above 60%, respectively.

### 2.3. Determining Tumor Methylation Status via MS-MLPA

DNA extracted from 29 tumor tissues was analyzed via the methylation-specific multiplex ligation-dependent probe amplification (MS-MLPA; MRC Holland, Amsterdam, The Netherlands) method, using a Probemix ME030-BWS/RSS (MRC Holland, Amsterdam, The Netherlands) in accordance with the manufacturer’s instructions. The methylation status at the imprinted 11p15 regions *H19* (ICR1) and *KvDMR* (ICR2) was determined via Coffalyser.Net software, version 220513.1739 [26]. The analysis was performed based on previously defined parameters. For H19, normal methylation rates were defined as 41–59%, a gain of methylation (GOM) was defined as >59%, and a loss of methylation (LOM) was defined as <40%. For KvDMR, normal methylation was considered >44%, and a LOM was defined as <44% [27].

### 2.4. Determining CfDNA Methylation Status via Digital PCR (dPCR)

Approximately 3 ng of cfDNA was subjected to bisulfite conversion using the EZ DNA Methylation Lightning Kit (cat D5031, Zymo Research, Irvine, CA, USA). Various concentrations of DNA, primers, and probes were tested at different temperatures and annealing conditions to determine the parameters for a digital PCR (dPCR) analysis. A 40 μL reaction mixture containing 1 ng of bisulfite-converted cfDNA, 200 nM of each of the forward and reverse primers, 100 nM of probes, and the Probe PCR Master Mix (Qiagen, Hilden, Germany; cat. ID: 250102) was prepared. The reaction was performed in 24-well nanoplates with 26k partitions and evaluated using QIAcuity One equipment (Qiagen, Hilden, Germany). The primer and probe sequences were obtained from a published study [28] and synthesized by Merck (St. Louis, MO, USA). The primer sequences were as follows: CTCF6F-GTATAGTATATGGGTATTTTTGGAGG and CTCF6R-CCCAATTAAAACRAACTCRAACTATAAT. The probe sequences were CTCF6M-AAGTGGTCGCGCGGCGGTAGTGTA, labeled with FAM for methylated DNA, and CTCF6U-TGGAAGTGGTTGTGTGGTGGTAGTGTAGG, labeled with HEX for unmethylated DNA. The cycling parameters were 95 °C for 2 min, followed by 50 cycles of 95 °C for 15 s, 52 °C for 30 s, and 60 °C for 1 min.

Data were analyzed using QIAcuity Software Suite, version 2.0.20 (SOW-975) (Qiagen, Hilden, Germany), and the thresholds were defined manually by considering the basal fluorescence given by the NTCs and the values obtained from the control sample, which comprised DNA extracted from the peripheral blood of normal individuals. The volume precision factor (VPF) was applied to enhance the precision of the concentration measurements. Imprinting at ICR1 was considered normal for values between 40% and 61%.

Figures were created with via BioRender.com, accessed on 8 August 2023.

## 3. Results

### 3.1. The Characterization of Methylation Status in ICR1 and ICR2, Located at the 11p15 Region in Embryonic Tumors

The methylation levels of the CpG sites located at ICR1 and ICR2 were used to conduct a hierarchical clustering analysis of the samples, using the Pearson correlation and complete linkage while maintaining the order of the CpG sites according to the chromosomal location. The clusterization was not associated with sex, clinical stage, metastasis at diagnosis, outcome (death), or age at diagnosis. The formation of these clusters was highly reproducible, with 95–100% bootstrap consistency (Figure 1A).

Cluster 1 presents a tendency for a loss of methylation (LOM) at ICR1 in at least part of the samples. These samples primarily consist of medulloblastomas belonging to molecular class groups 3 and 4. A similar analysis of the methylation data from an independent group of 36 medulloblastomas [29] reported the formation of three clusters: one resembling the pattern observed in cluster 1 (ICR1 LOM), one similar to cluster 3.1 (both ICR1 GOM and ICR2 LOM), and a third cluster characterized by ICR1 LOM and ICR2 GOM. Nine medulloblastomas had 11p15 LOM and/or GOM (Figure 1B).

Cluster 2 comprises mostly samples with normal ICR1 and ICR2 methylation levels. Samples from cluster 3 are characterized by ICR1 GOM: two hepatoblastomas (6.2%), three medulloblastomas (9.4%), seven neuroblastomas (21.9%), four embryonal rhabdomyosarcomas (12.5%), one rhabdomyosarcoma (3.1%), one glioblastoma (3.1%), and fourteen Wilms tumors (43.7%). Within cluster 3, a subcluster labeled cluster 3.1 exhibited both ICR1 GOM and ICR2 LOM; it included seven Wilms tumors, three embryonal rhabdomyosarcomas, one neuroblastoma, two hepatoblastomas, and one medulloblastoma. Heterogeneous levels of ICR1 LOI were observed in cluster 3.3, which consisted of two Wilms tumors, five neuroblastomas, one rhabdomyosarcoma, one glioblastoma, and two medulloblastomas. Among the 99 embryonic tumors analyzed, 46 (46.5%) exhibited ICR1 LOM and/or GOM (Table 1).

These findings highlight the presence of differential methylation patterns at ICR1 and ICR2 in embryonic tumors, particularly in Wilms tumors and neuroblastomas, with a significant proportion showing ICR1 methylation level changes.

Out of 99 cases, 29 (29.3%) were selected for further analysis as these participants had peripheral blood samples collected at the time of diagnosis, prior to any therapeutic intervention. These samples were assayed via MS-MLPA to confirm their methylation status. Consistent with the clustering analysis, a significant proportion of Wilms tumors (81.8%) exhibited ICR1 GOM, as did 30.4% of neuroblastomas. Some samples of Wilms tumors, neuroblastomas, embryonal rhabdomyosarcomas, and hepatoblastomas also had ICR2 LOM, and no alterations in methylation levels were observed at either ICR1 or ICR2 in Ewing sarcomas. Overall, 51.7% of the 29 embryonic tumors showed ICR1 GOM, while 37.9% exhibited ICR2 LOM (Figure 2).

Regarding copy number alterations analyzed via MS-MLPA, most cases did not show significant changes. Two tumors (one neuroblastoma and one embryonal sarcoma) demonstrated ICR1 gain, two tumors (one embryonal sarcoma and one Wilms tumor) demonstrated ICR1 loss, and one Wilms tumor demonstrated ICR2 gain (Figure 1C).

### 3.2. ICR1 Methylation Status Is Detectable at the cfDNA at Diagnosis

The majority of samples (76%) showed agreement between cfDNA detection via dPCR and tumor samples analyzed via MS-MLPA (Figure 2, Appendix A). By comparing the technical results obtained from the EPIC Beadchip arrays and MS-MLPA, we found high values of sensibility and specificity. Furthermore, when considering the source of DNA (tumor or cfDNA), 22 cases (76%) showed agreement between tumor DNA and cfDNA, with a sensitivity of 86.7% and a specificity of 64.3% (Table 2), suggesting that the detection of ICR1 methylation status in cfDNA via dPCR is consistent with the methylation status observed in tumor DNA.

Among the disagreements, three samples, ES-30, NB-26, and WT-14, were located in cluster 3.3, which is characterized by some heterogeneity with respect to the ICR1 methylation level. Considering the challenge of characterizing tumor heterogeneity, the cfDNA dPCR seems to overcome the MS-MLPA limit and identify ICR1 methylation changes, as indicated by the EPIC Beadchip arrays.

For six cases, we also evaluated the ICR1 status at one to three points during follow-up; however, there was no association between ICR1 status and outcome during the period analyzed (Figure 3). Nevertheless, these patients will continue to be followed for a longer period before we discard the data.

## 4. Discussion

Embryonic tumors, despite having few recurrent single-nucleotide variants (SNVs), are characterized by broad epigenetic alterations, a reflex of their origin which is closely related to an impairment during cell differentiation [30,31,32,33]. The methylomes of these tumors can provide insights into their cells of origin and potentially predict a patient’s outcome, a characteristic that has been used for tumor classification [34,35]. Among the epigenetic modifications, alterations in 11p15 are observed as germline alterations in children with Beckwith–Wiedemann syndrome, resulting in an increased risk of developing cancer. Approximately 8% of these children develop embryonic tumors, with the most common types being Wilms tumor (52% of all tumors), hepatoblastomas (14%), neuroblastomas (10%), and rhabdomyosarcomas (5%) [12].

We retrieved the methylation status of ICR1 and ICR2 at 11p15 in a group of 99 sporadic embryonic tumors that were consecutively assayed at our laboratory. 11p15 is an imprinted region which results in the monoallelic expression of a subset of genes, depending on their parental origin [36]. Around half of the tumors (51.5%) had an alteration in the methylation levels of 11p15 as a result of ICR1 LOM and/or GOM or ICR2 LOM. Medulloblastomas (32.6%), Wilms tumors (30.4%) and neuroblastomas (15.2%) constituted the most frequently identified tumor types. These findings have been reported in the literature, especially for Wilms tumors [10,11,37]. Also, these tumors are among the most frequently found tumors in patients with Beckwith–Wiedmann syndrome, who present alterations in 11p15 as the underlying germinative alteration [9]. As a consequence of being found in several types of embryonic tumors, 11p15 alterations cannot be used for differential diagnosis.

The hierarchical clustering based on the methylation levels of ICR1 and ICR2 discriminated four groups, with nearly half (46%) presenting some level of methylation change. The profiles suggest that the mechanisms associated with the development of Wilms tumors and medulloblastomas are distinct, either by location (ICR1 or ICR2) or by the direction of the methylation (gain or loss). Given that most samples with ICR1 GOM were Wilms tumors and those with ICR2 LOM were hepatoblastomas, one can hypothesize that the regulation of ICR1 methylation is involved in kidney cell differentiation, whilst the regulation of ICR2 is involved in liver cell differentiation. This hypothesis is supported by the fact that patients with Beckwith–Wiedemann syndrome with Wilms tumors seem to have a tendency to have ICR1 GOM, and the development of hepatoblastomas seems to be related to LOI due to uniparental disomy [38].

Alterations in 11p15 in Wilms tumors are the result of either LOI or LOH, which are found in 70% of cases, independent of tumor histology [39]. We reported that 82.4% of Wilms tumors present 11p15 alterations, with a subgroup presenting alteration at both ICR1 and ICR2. Cases with both ICR1 GOM and ICR2 LOM are suggestive of LOH (the loss of maternal copy) or paternal UPD at 11p15.5 [4,12]. All cases were unilateral and sporadic, and we did not find relationships with other clinical and pathological characteristics; this could be related to the small number of cases in this study, which is one of its limitations. We only analyzed two clear cell sarcomas of the kidney; none presented alterations at 11p15, but one case (CCSK-38) demonstrated ICR1 GOM, which was detected in the liquid biopsy assay. In a group of 30 cases of clear cell sarcomas of the kidney, 43% had *IGF2* overexpressed considered *IGF2* LOI, with retention of normal *H19* expression [40]. Considering the advantage of dPCR in capturing the ICR1 LOI, as we showed, it goes against a refined diagnostic tool as both renal tumors (Wilms tumor and clear cell sarcoma of the kidney) may present this alteration.

All 99 embryonal tumors were tested via an EPIC array. Of these tumors, 29 were also accessed via MS-MLPA and cfDNA (Figure 2). Of the 29 cases tested for cfDNA, 6 had blood collected at diagnosis and follow-up which were used to test ICR1 methylation status as biomarker for monitoring disease progression. Our findings demonstrate good agreement between the results obtained from the tumor samples analyzed via EPIC Beadchip arrays and MS-MLPA. Moreover, the dPCR method was shown to be robust as it correctly identified the presence or absence of ICR1 methylation in cfDNA samples in peripheral blood at the moment of diagnosis. Moreover, our findings suggest that the analysis of cfDNA via dPCR has the potential to overcome the limitations of MS-MLPA in detecting ICR1 methylation status, similar to what was identified via the EPIC Beadchip arrays. Tumor heterogeneity can pose challenges in accurately characterizing the methylation status of specific regions, and it appears that assessing cfDNA via dPCR may provide a more comprehensive and representative assessment of ICR1 methylation status. Herein, we analyzed tumor tissues but not peripheral blood cells, which prevented us from excluding children with alterations in mosaic or in the Beckwith–Wiedemann syndrome spectrum that did not have a clinical presentation, which is a limitation of this study. However, if present, these alterations would be identified in cfDNA methylation experiments.

Although the dPCR is a promising tool for improving pediatric cancer diagnosis, risk stratification, and the tracking of treatment response and disease recurrence [41], the characterization of ICR1 methylation status during a patient’s follow-up did not correlate with outcome. Other epigenetic alterations, such as *RASSF1A* hypermethylation [42], might be more noteworthy.

MS-MLPA is considered a gold standard methodology for the molecular diagnosis of several diseases caused by abnormal DNA CNV and methylation [43]. When comparing this methodology with EPIC Beadchip arrays, it was found that the MS-MLPA could not capture tumor heterogeneity. However, the MS-MLPA method used in this study was developed to assay peripheral blood, searching for germline alterations, which contain no or low levels of heterogeneity. These technical disagreements are exemplified by two rhabdomyosarcomas that had ICR2 LOM: RBM-33 demonstrated a loss of methylation on the CpG site, where there was an MS-MLPA probe, which was thus detected, while RBM-36 also had a tendency toward losses of methylation in CpG sites different from the ones assayed via MS-MLPA probes; thus, the LOM was not detected. Nevertheless, the MS-MLPA provides additional confirmation of the methylation status in the selected cases, with a substantial proportion of Wilms tumors and neuroblastomas showing changes in methylation levels at ICR1 and ICR2. An analysis of copy number alterations revealed relatively low frequencies of ICR1 gain, ICR1 loss, and ICR2 gain in the examined samples. These findings indicate that CNV events involving 11p15 are not the most common mechanism associated with methylation alterations in these sporadic embryonic tumors.

## 5. Conclusions

This study highlights the prevalence of 11p15 LOI in sporadic embryonic tumors, with differences relating to methylation levels (gain or loss), location (ICR1 or ICR2), and tumor types (medulloblastomas, Wilms tumor, and hepatoblastomas). Around half of the tumors (51.5%) had an 11p15 LOI as a result of ICR1 LOM and/or GOM or ICR2 LOM. Among these, the more frequent tumors comprised medulloblastomas (32.6%), Wilms tumors (30.4%), and neuroblastomas (15.2%). Our findings demonstrate good agreement between the results obtained from tumor samples analyzed via EPIC Beadchip arrays and MS-MLPA and also between tumor tissue and cfDNA.

## Figures and Tables

**Figure 1 cancers-15-04256-f001:**
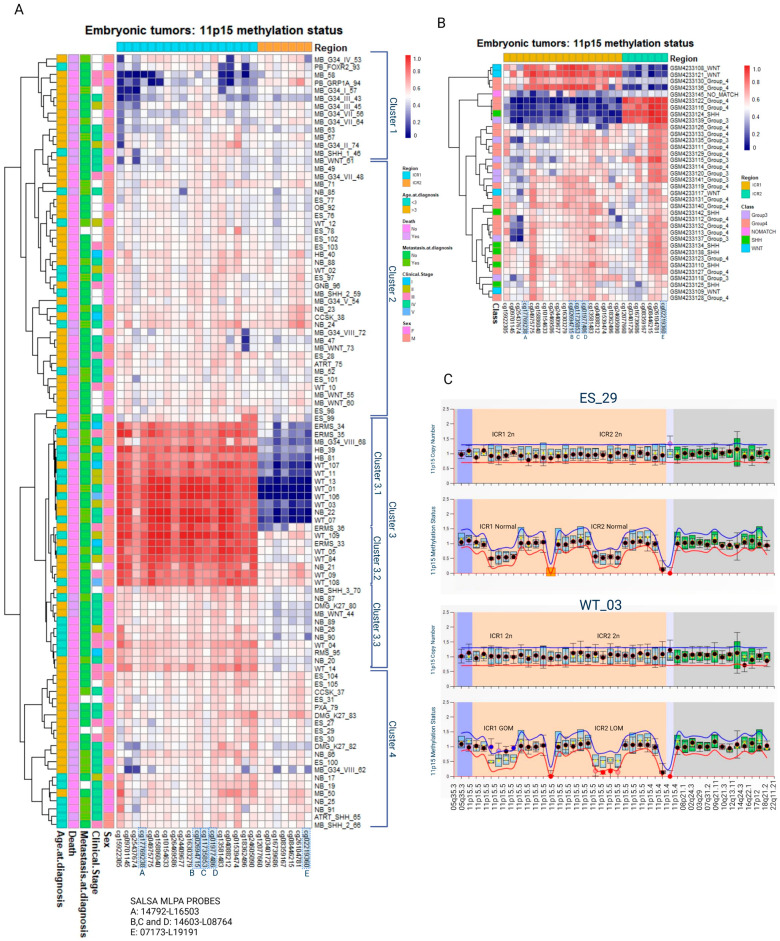
Methylation levels of CpG sites located at either ICR1 or ICR2 (11p15) were retrieved from embryonic tumors assessed via EPIC Beadchip arrays. ICR1-Ch11:2019379-2020007 are represented by 17 CpGs. ICR2-Ch11: 2720540-2721394 are represented by 7 CpGs. CpG sites highlighted in blue correspond to CpG sites also analyzed via MS-MLPA. Information related to clinical stage, sex, death, metastasis, and age at diagnosis are displayed on the left side. (**A**) Hierarchical clusterization of 99 embryonic tumors and (**B**) 36 medulloblastomas extracted from GSE142627. (**C**) Representative MS-MLPAs showing ICR1 GOM and ICR2 LOM. WT, Wilms tumor; NB, neuroblastoma; ES, Ewing sarcoma; ERMS, embryonal rhabdomyosarcoma; RMS, rhabdomyosarcoma; CCSK, clear cell sarcoma of the kidney; HB, hepatoblastoma; MB, medulloblastoma; GB, glioblastoma; OB, osteoblastoma; PB, pineoblastoma; GNB, ganglioneuroblastoma; ATRT, Atypical Teratoid Rhabdoid Tumor; GOM, gain of methylation; LOM, loss of methylation; M, male; F, female.

**Figure 2 cancers-15-04256-f002:**
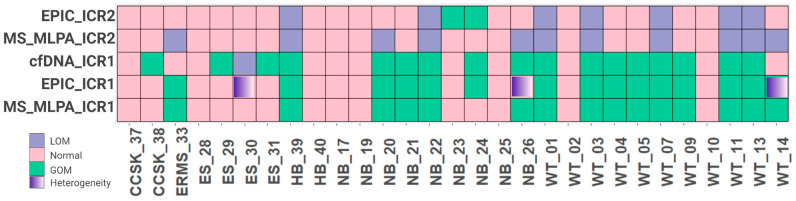
Comparison of 29 samples accessed via EPIC array, MS-MLPA, and cfDNA. WT, Wilms tumor; NB, neuroblastoma; ES, Ewing sarcoma; ERMS, embryonal rhabdomyosarcoma; RMS, rhabdomyosarcoma; CCSK, clear cell sarcoma of the kidney; HB, hepatoblastoma; GOM, gain of methylation; LOI, loss of imprinting.

**Figure 3 cancers-15-04256-f003:**
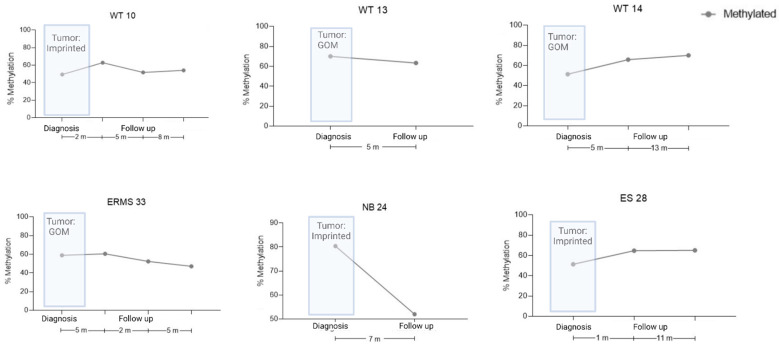
cfDNA levels during the follow-up of six patients, as measured by ICR1 status via dPCR. Tumor ICR1 status is shown at diagnosis, as measured via MS-MLPA. Methylation percentage in cfDNA patients’ samples during clinical events. WT, Wilms tumor; NB, neuroblastoma; ES, Ewing sarcoma; ERMS, embryonal rhabdomyosarcoma.

**Table 1 cancers-15-04256-t001:** Tumors with alterations in the methylation levels of ICR1 and/or ICR2 (11p15).

Tumor Type	Characterized Tumors (n)	Tumors with ICR1 LOM	Tumors with ICR1 GOM	Tumors with ICR2 LOM
Atypical teratoid rhabdoid tumor	2	0	0	0
Clear cell sarcoma of the kidney	2	0	0	0
Diffuse midline gliomas H3 K27-altered *	3	0	1 (3.3%)	0
Embryonal rhabdomyosarcoma	4	0	4 (100%)	3 (75%)
Ewing sarcoma	17	0	0	0
Ganglioneuroblastoma	1	0	0	0
Hepatoblastoma	3	0	2 (67.3%)	2 (67.3%)
Medulloblastoma, unclassified	8	3 (37.5%)	0	0
Medulloblastoma_G34 *	12	7 (58.3%)	1 (8.33%)	1 (8.33%)
Medulloblastoma_SHH *	4	1 (25%)	1 (25%)	0
Medulloblastoma_WNT *	5	1 (20%)	1 (20%)	0
Neuroblastoma	16	0	7 (43.75%)	1 (6.25%)
Osteoblastoma	1	0	0	0
Pineoblastoma_FOXR2 *	1	1 (100%)	0	0
Pineoblastoma_GRP1A *	1	1 (100%)	0	0
Pleomorphic xanthoastrocytoma *	1	0	0	0
Rhabdomyosarcoma	1	0	1 (100%)	0
Wilms tumor	17	0	14 (82.35%)	7 (41.17%)

* The WHO 2021 classification was used for central nervous system tumors.

**Table 2 cancers-15-04256-t002:** Sensibility and specificity from technical (EPIC and MS-MLPA) and biological (tumor tissue and cfDNA) experiments.

Comparison	Sensitivity (%)	Specificity (%)	Positive Predictive Value (%)	Negative Predictive Value (%)
ICR1 Technical (EPIC/MS-MLPA)	93.3	92.9	93.3	92.9
ICR2 Technical (EPIC/MS-MLPA)	63.6	88.9	77.8	80
ICR1 Biological (MS-MLPA/cfDNA)	86.7	64.3	72.2	81.8

## Data Availability

All data used in this study are included in the manuscript or the Appendix A. Raw data from the cfDNA experiments can be shared upon request.

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
