# Peer review of "11p15 Epimutations in Pediatric Embryonic Tumors: Insights from a Methylome Analysis"

_cancers, 2023, doi:10.3390/cancers15174256_

Round 1
Reviewer 1 Report
The authors investigated the occurrence of aberrant DNA methylation and LOI as a common characteristic of embryonic tumours.
They present compelling evidence of the prevalence of 11p15 LOI in sporadic embryonic tumors, 294 with differences related to methylation levels (gain or loss), location (ICR1 or ICR2) and 295 between tumor types (medulloblastoma, Wilms tumor and hepatoblastoma) and demonstrate good correlation between EPIC Beadchip arrays and MS-MLPA.
I would recommend a revision of the paper in terms of clarity. A review by an English native speaker could help.
In addition, I would recommend the revision of the simple summary. If this is supposed to serve as a lay summary, the language used is still too technical for a lay public.
No additional comments
Author Response
Thank you for the careful review of the manuscript and the questions raised by the reviewers. The comments improved the manuscript and we hope it is now suitable for publication.
Regarding your questions:
I would recommend a revision of the paper in terms of clarity. A review by an English native speaker could help.
The full paper was reviewed and corrected.
I would recommend the revision of the simple summary. If this is supposed to serve as a lay summary, the language used is still too technical for a lay public.
As suggested, the simple summary was reviewed and text was adapted.
Reviewer 2 Report
Torres Silva and colleagues analyzed 99 non-syndromic embryonic tumors to study DNA methylation at the two imprinted regions ICR1 and ICR2, using the EPIC methylation array. Some defects were confirmed by MS-MLPA in tumors and by digital PCR in circulating tumoral cell-free DNA.
I have some comments:
- I disagree with the title and with the use the authors did of the term ‘Loss of Imprinting’ (LOI). LOI refers to the allelic gene expression, that the authors do not test at all. There is LOI when an imprinted gene becomes biallelically activated or biallelically silenced. In this manuscript, the authors analyzed DNA methylation at two imprinted loci and identified epigenetic abnormalities that only suggest LOI but do not demonstrate it. Better just use the terms LOM for Loss of Methylation and GOM for Gain of Methylation, instead of LOI.
- The authors did not use normal controls in the EPIC array analysis. Please clarify how DNA methylation alterations were identified at both ICR1 and ICR2. Which was the bioinformatic strategy to define normal methylation, LOM and GOM?
- Part of the selected cohort of embryonic tumors (such as Wilms tumors, hepatoblastoma and rhabdomyosarcoma) are associated with Beckwith-Wiedemann syndrome (BWS), an imprinting disorder known to have methylation defects at ICR1 and/or ICR2. What about the other tumor types? Is it known their association with epigenetic alteration at the 11p15.5 region? A better description of this point in the Introduction is relevant to understand the novelty of the study.
- To better follow the results, please make an additional table with the DNA methylation levels of ICR1 and ICR2 for each tumor analyzed by different approaches: in tumor tissues (for a comparison EPIC array and MS-MLPA) and in the cfDNA.
- In the Abstract (page 1, lines 25-26), the authors report ICR1 GOM in medulloblastoma and ICR1 LOM in Wilms tumors and hepatoblastoma but results in Fig.1A indicate the contrary.
- In the Abstract (page 1, lines 22), instead of tumors ‘not related to this syndrome’, better to say ‘non-syndromic’ or ‘isolated’ tumors.
- In the Discussion (page 7, line 236) and in the Conclusions (page 8, line 297), the authors refer to samples with ICR2 GOM. It’s not clear to me what are these tumors. Please explain better.
- In the Discussion, the authors should mention that the tumors with both ICR1 GOM and ICR2 LOM are cases suggestive of LOH (loss of maternal copy) or paternal UPD at chr 11p15.5.
- Again in the Discussion (page 8, lines 277-279), please clarify that the applications of MS-MLPA are both CNV and DNA methylation analyses.
- In Fig.2, please indicate LOM instead of LOI.
- In Fig.1A and B the two regions are indicated as IC1 and IC2. Please use a uniform nomenclature.
- Also circulating cell-free DNA is sometimes indicated as cfDNA and some others as ctDNA. Again, please use a uniform nomenclature.
- May the authors indicate in Fig.1A, as they did in Fig.1B, the CpG sites also analyzed by MS-MLPA? Moreover, Fig.1 should take one entire page to make heatmaps bigger and clearer.
- At page 2, line 46: ICRs have differential DNA methylation between ‘alleles’, not ‘strands’.
- Are genomic coordinates at page 2, lines 94-95, referred to GRCh37/hg19? Please indicate.
- At page 4, line 160, better to say: Cluster 2 comprises mostly samples with normal ICR1 and ICR2 methylation.
Author Response
Thank you for the careful review of the manuscript and the questions raised by the reviewers. The comments improved the manuscript and we hope it is now suitable for publication.
Regarding your questions:
- I disagree with the title and with the use the authors did of the term ‘Loss of Imprinting’ (LOI). LOI refers to the allelic gene expression, that the authors do not test at all. There is LOI when an imprinted gene becomes biallelically activated or biallelically silenced. In this manuscript, the authors analyzed DNA methylation at two imprinted loci and identified epigenetic abnormalities that only suggest LOI but do not demonstrate it. Better just use the terms LOM for Loss of Methylation and GOM for Gain of Methylation, instead of LOI.
The title was changed accordingly. The text was reviewed and LOI was changed to LOM or GOM when appropriated.
- The authors did not use normal controls in the EPIC array analysis. Please clarify how DNA methylation alterations were identified at both ICR1 and ICR2. Which was the bioinformatic strategy to define normal methylation, LOM and GOM?
We included this information in the methods section (tumor methylation status by EPIC Beadchip arrays).
- Part of the selected cohort of embryonic tumors (such as Wilms tumors, hepatoblastoma and rhabdomyosarcoma) are associated with Beckwith-Wiedemann syndrome (BWS), an imprinting disorder known to have methylation defects at ICR1 and/or ICR2. What about the other tumor types? Is it known their association with epigenetic alteration at the 11p15.5 region? A better description of this point in the Introduction is relevant to understand the novelty of the study.
This information was included in the introduction. It is worth to note that due to the rarity of the embryonic tumors, very few studies other than some case reports have characterized 11p15 status in embryonic tumors. The exceptions are the ones already included in the Introduction.
- To better follow the results, please make an additional table with the DNA methylation levels of ICR1 and ICR2 for each tumor analyzed by different approaches: in tumor tissues (for a comparison EPIC array and MS-MLPA) and in the cfDNA.
All 99 embryonic tumors were tested by EPIC array. Of this, 29 were also accessed by MS-MLPA and cfDNA (cases described in Figure 2). Of 29 cases tested for cfDNA (ICR1), 6 also had blood collected at follow up and ICR1 methylation status. The table was included as a supplemental file.
- In the Abstract (page 1, lines 25-26), the authors report ICR1 GOM in medulloblastoma and ICR1 LOM in Wilms tumors and hepatoblastoma but results in Fig.1A indicate the contrary.
Thank you for pointing this out. This result was corrected in the abstract, which is now in agreement with figure 1A
- In the Abstract (page 1, lines 22), instead of tumors ‘not related to this syndrome’, better to say ‘non-syndromic’ or ‘isolated’ tumors.
We agreed with this comment and changed it to “sporadic tumors”.
- In the Discussion (page 7, line 236) and in the Conclusions (page 8, line 297), the authors refer to samples with ICR2 GOM. It’s not clear to me what are these tumors. Please explain better.
Thank you for pointing this out. We reported LOM at ICR2, but not GOM (Figure 1A), this was corrected in the discussion and conclusion sections. However, when analyzing a higher number of medulloblastomas, made available by another study, it was possible to identify a more clear pattern and the addition change of GOM at ICR2 (Figure 1B). We believe that the characterization of additional cases in the literature as a whole may provide a clearer picture in the future.
- In the Discussion, the authors should mention that the tumors with both ICR1 GOM and ICR2 LOM are cases suggestive of LOH (loss of maternal copy) or paternal UPD at chr 11p15.5.
We added this information in the discussion section
- Again in the Discussion (page 8, lines 277-279), please clarify that the applications of MS-MLPA are both CNV and DNA methylation analyses.
This information was incorporated in the discussion
- In Fig.2, please indicate LOM instead of LOI.
As suggested, LOM was indicated in Figure 2.
- In Fig.1A and B the two regions are indicated as IC1 and IC2. Please use a uniform nomenclature.
Thanks for this correction. ICR1 and ICR2 were indicated in figure 1A and B. The original text was also verified for a uniform nomenclature.
- Also circulating cell-free DNA is sometimes indicated as cfDNA and some others as ctDNA. Again, please use a uniform nomenclature.
Here, we highlight the difference between cfDNA referred to as all amount of DNA present in plasma, and ctDNA, when an alteration (LOM or GOM in this case) was identified.
- May the authors indicate in Fig.1A, as they did in Fig.1B, the CpG sites also analyzed by MS-MLPA? Moreover, Fig.1 should take one entire page to make heatmaps bigger and clearer.
CpG sites represented in the MS-MLPA were also indicated in Figure 1B. We agree that Figure 1 could be bigger for better visualization. They were submitted as separated files (high-resolution) for the Journal.
- At page 2, line 46: ICRs have differential DNA methylation between ‘alleles’, not ‘strands’.
We altered this as suggested
- Are genomic coordinates at page 2, lines 94-95, referred to GRCh37/hg19? Please indicate.
This information is stated in the Methods section (“... ICR1 (GRCh37/hg19 chr11:2019379-2020007) and ICR2 (GRCh37/hg19 chr11:2720540-2721394)...”)
- At page 4, line 160, better to say: Cluster 2 comprises mostly samples with normal ICR1 and ICR2
We altered this as suggested.
Reviewer 3 Report
Very interesting scientific work. The authors compare the results of DNA methylation studies of three methods. (Epic array, MS_MLPA, and dPCR on cfDNA). The results of the research will certainly be useful when planning further studies on the methylation of embryonic tumors. However, I believe that the work requires major and minor corrections.
Major points to improve:
1. Line 45-58: I gently suggest expanding this part of the introduction. There are differences in risk of developing cancer depending on the mechanism of methylation change in ICR1 and ICR2. Moreover, not every LOI results in gain of methylation. This part of the introduction is too simplistic and therefore not entirely correct.
Relevant recent literature on epigentic studies in embryonic tumors is missing (et al 2022, Fiala et al, 2023 and others).
2, The aim of the study is not clear. Is it the validation/comparison of the techniques used (technical part is fine!), is it the determination of the methylarion status at 11p15 or is it the early detection of methylation changes in cfDNA?
3. The analysis are perfomed on sporadic tumors of patients that were not previously diagnosed with BWS. The methylation changes found are specifically associated with BWS, maybe the authors can comment on that or show clinical data on the patients (table in suppl. data?).
4. It should be very clear which patients were tested by Epic array (99), MS-MLPA (29?) and tumor tissue? ) and cfDNA(?). A table summarizing/comparing the test results of each patient by each method may be helpful here.
The text contains grammatical, stylistic and punctuation errors. In some sentences the words are wrongly chosen.
Author Response
Thank you for the careful review of the manuscript and the questions raised by the reviewers. The comments improved the manuscript and we hope it is now suitable for publication.
Regarding your questions:
Extensive editing of English language required.
The text was reviewed.
Very interesting scientific work. The authors compare the results of DNA methylation studies of three methods. (Epic array, MS_MLPA, and dPCR on cfDNA). The results of the research will certainly be useful when planning further studies on the methylation of embryonic tumors. However, I believe that the work requires major and minor corrections.
We thank the comment.
Major points to improve:
1. Line 45-58: I gently suggest expanding this part of the introduction. There are differences in risk of developing cancer depending on the mechanism of methylation change in ICR1 and ICR2. Moreover, not every LOI results in gain of methylation. This part of the introduction is too simplistic and therefore not entirely correct. Relevant recent literature on epigentic studies in embryonic tumors is missing (Stoltze UK et al 2022, Fiala et al, 2023 and others).
Information regarding the differences between tumor types related to ICR1 or ICR2 is included in the discussion section. We checked the literature again for relevant studies and included it in the introduction and discussion sections.
2. The aim of the study is not clear. Is it the validation/comparison of the techniques used (technical part is fine!), is it the determination of the methylation status at 11p15 or is it the early detection of methylation changes in cfDNA?
In this study, we first aimed to characterize the 11p15 status to gain insights of the methylation pattern in embryonic tumors, including entities that lack this information in the literature. We also tested ICR1 LOI as a biomarker to be used as liquid biopsy although, with these data, we concluded that it will not add information to help diagnosis or disease monitoring (as discussed).
3. The analysis are perfomed on sporadic tumors of patients that were not previously diagnosed with BWS. The methylation changes found are specifically associated with BWS, maybe the authors can comment on that or show clinical data on the patients (table in suppl. data?).
The germline 11p15 LOI is associated with Beckwith-Wiedmann syndrome but at somatic level, it may disrupt in any type of cancer. The link between the tumors we characterized is that they are all classified as an embryonic tumor. Embryonic tumors are characterized by a histological component of small blue round cells and morphologically resemble the fetal organ of origin in contrast to the fully differentiated organ. Due to the relatively small number of mutations in recurrent genes, studies characterizing the epigenetic mechanisms changes have been showing that DNA methylation changes have a prominent role in these tumors, regardless of being sporadic or associated with a syndrome. It is also of note that Beckwith-Wiedmann syndrome is very rare (1:11,000 births) and may be not fully characterized.
4. It should be very clear which patients were tested by Epic array (99), MS-MLPA (29?) and tumor tissue? ) and cfDNA(?). A table summarizing/comparing the test results of each patient by each method may be helpful here.
All 99 embryonic tumors were tested by EPIC array. Of this, 29 were also accessed by MS-MLPA and cfDNA (cases described in Figure 2). Of 29 cases tested for cfDNA (ICR1), 6 also had blood collected at follow up ICR1 methylation status. The table was included as a supplemental file.
Round 2
Reviewer 2 Report
Torres Silva and colleagues replied to most of my observations. However, I still have some comments:
- In the Introduction, line 63, the sentence “LOI at 11p15 is the germline epimutation associated with Beckwith-Wiedemann Syndrome (BWS)…” is not correct. LOI is not an epimutation. In BWS, LOI is due to epimutations at 11p15.5. Moreover, neither epimutations nor LOI are germline derived in BWS.
- In the Results, line 173, again there is a mistake relative to cluster 3, characterized by ICR1 “GOM”, not “LOM”.
- Figure 1 is still too small to be visible and clear. Please adjust it to a full A4 page in size.
- In the Discussion, line 253, the sentence “LOI is observed as a germline alteration in children with Beckwith-Wiedemann syndrome…” is not true.
- In the Discussion, line 263, neuroblastoma is repeated two times. Same in the Conclusions, lines 334-335.
Author Response
We thank this reviewer for the careful evaluation of the manuscript. Changes are highlighted in blue in the manuscript and below are the responses to the comments.
Torres Silva and colleagues replied to most of my observations. However, I still have some comments:
- In the Introduction, line 63, the sentence “LOI at 11p15 is the germline epimutation associated with Beckwith-Wiedemann Syndrome (BWS)…” is not correct. LOI is not an epimutation. In BWS, LOI is due to epimutations at 11p15.5. Moreover, neither epimutations nor LOI are germline derived in BWS.
The text was re-write to answer to this comment.
- In the Results, line 173, again there is a mistake relative to cluster 3, characterized by ICR1 “GOM”, not “LOM”.
Thank you. It has been corrected.
- Figure 1 is still too small to be visible and clear. Please adjust it to a full A4 page in size.
We changed again but we ask for the editor to consider the full size image, uploaded separately to prepare the final version of the manuscript.
- In the Discussion, line 253, the sentence “LOI is observed as a germline alteration in children with Beckwith-Wiedemann syndrome…” is not true.
Thank you. It has been corrected.
- In the Discussion, line 263, neuroblastoma is repeated two times. Same in the Conclusions, lines 334-335.
Thank you. It has been corrected.
Reviewer 3 Report
My first problem with the paper was that the aim of the study was not really clear in the first version. Although there are certainly improvements in that respect, so now I better understand what the idea behind this paper is.
My second problem was that although the authors claim that these are sporadic cases, the methylation changes detected are specific for BWS syndrome patients. Since there are no methylation data of the patients in blood it can not be excluded that these are BWSspectrum patients that do show less features then classical BWS patients. If any of these patients show loss of methylation in blood, the conclusions of the authors may not be correct. Findings in cfDNA might be due to a congenital methylation change. That would also be another explanation of the findings in the follow up study in 6 patients.
In table 1 it is unclear which patients have pUPD11p15. In the number they are (I think) counted as GOM IC1 and LOM IC2, both true methylation disturbances. pUPD11p15 is a chromosomal aberration, and should be viewed as a separate entity.
And still relevant literature is missing ie on molecular findings in tumours associated with BWS such as Wilm's tumour.
Author Response
R3: My first problem with the paper was that the aim of the study was not really clear in the first version. Although there are certainly improvements in that respect, so now I better understand what the idea behind this paper is.
Reviewer 3 seems satisfied with the modifications we did.
R3: My second problem was that although the authors claim that these are sporadic cases, the methylation changes detected are specific for BWS syndrome patients. Since there are no methylation data of the patients in blood it can not be excluded that these are BWSspectrum patients that do show less features then classical BWS patients. If any of these patients show loss of methylation in blood, the conclusions of the authors may not be correct. Findings in cfDNA might be due to a congenital methylation change. That would also be another explanation of the findings in the follow up study in 6 patients.
We agree with this comment and included this discussion as a limitation of the study.
R3: In table 1 it is unclear which patients have pUPD11p15. In the number they are (I think) counted as GOM IC1 and LOM IC2, both true methylation disturbances. pUPD11p15 is a chromosomal aberration, and should be viewed as a separate entity.
Table 1 shows data from the EPIC Beadchip array experiments which evaluate DNA methylation and copy number but not uniparental disomy. That's the reason we cannot include this information in the table.
R3: And still relevant literature is missing ie on molecular findings in tumours associated with BWS such as Wilm's tumour."
We understand that 11p15 alterations were explored in Wilms tumor, which is reflected by the 9 references that directly cite Wilms tumors, and some others that evaluated Wilms tumors with others. Although we did not cite all the studies that covered the aspects cited by this reviewer, we believe that the information in the manuscript is properly referenced.